# Causal graph analysis of COVID-19 observational data in German districts reveals effects of determining factors on reported case numbers

**Edgar Steiger**\*, **Tobias Mussgnug**, **Lars Eric Kroll**

Central Research Institute of Ambulatory Health Care in Germany (Zi), Berlin, Germany

\* esteiger@zi.de

## Abstract

Several determinants are suspected to be causal drivers for new cases of COVID-19 infection. Correcting for possible confounders, we estimated the effects of the most prominent determining factors on reported case numbers. To this end, we used a directed acyclic graph (DAG) as a graphical representation of the hypothesized causal effects of the determinants on new reported cases of COVID-19. Based on this, we computed valid adjustment sets of the possible confounding factors. We collected data for Germany from publicly available sources (e.g. Robert Koch Institute, Germany's National Meteorological Service, Google) for 401 German districts over the period of 15 February to 8 July 2020, and estimated total causal effects based on our DAG analysis by negative binomial regression. Our analysis revealed favorable effects of increasing temperature, increased public mobility for essential shopping (grocery and pharmacy) or within residential areas, and awareness measured by COVID-19 burden, all of them reducing the outcome of newly reported COVID-19 cases. Conversely, we saw adverse effects leading to an increase in new COVID-19 cases for public mobility in retail and recreational areas or workplaces, awareness measured by searches for "corona" in Google, higher rainfall, and some socio-demographic factors. Non-pharmaceutical interventions were found to be effective in reducing case numbers. This comprehensive causal graph analysis of a variety of determinants affecting COVID-19 progression gives strong evidence for the driving forces of mobility, public awareness, and temperature, whose implications need to be taken into account for future decisions regarding pandemic management.

## Introduction

As the COVID-19 pandemic progresses, research on mechanisms behind the transmission of SARS-CoV-2 shows conflicting evidence [1–3]. While effects of mobility have been extensively discussed, less is known on other factors such as changing awareness in the population [4–6] or the effects of temperature [7–9]. A limiting factor in many studies is the lack of a causal

**Data Availability Statement:** All relevant data are available from public sources and are aggregated in the github repository pertaining to the

manuscript: https://github.com/zidatalab/causalcovid19.

**Funding:** The authors received no specific funding for this work.

**Competing interests:** The authors have declared that no competing interests exist.

approach to assess the causal contributions of various factors [10]. This can lead to distorted estimates of the causal factors with observational data [10–12].

With COVID-19, we find ourselves in a situation in which information on the causal contribution of various influencing factors in the population is urgently needed to inform politicians and health authorities. On the other hand, trials cannot be carried out for obvious ethical and legal reasons. Therefore, when assessing the effects of determinants of SARS-CoV-2 spread, special attention must be paid to strategies for the selection of confounding factors.

Another problem with assessing the effects of various determinants of SARS-CoV-2 spread is the heterogeneity of the countries and regions examined for example in the Johns Hopkins University (JHU) COVID-19 database [13]. The comparison of time series of case numbers from different countries and observational periods can be strongly distorted by different factors like testing capacities and regional variations.

Our objective is to provide valid estimates of the effects of the main drivers of the pandemic with a causal graph approach. We conducted a scoping review of the available studies regarding signaling pathways and determinants of the spread of SARS-CoV-2 infections and the reported new COVID-19 cases. Then we integrated the current findings into a directed acyclic graph for the progress of the pandemic at the regional level. Using the resulting model and the do-calculus we found identifiable effects without blocked causal paths whose effects can be analyzed with observational data [14]. We used regional time series data of all German districts (401) from various publicly available sources to analyze these questions on a regional level. Germany is a good choice in this regard, because it has ample data on contributing factors on the regional level and has had high testing and treatment capacities from early on in the pandemic.

## Causal model

We used a directed acyclic graph (DAG) [11, 12] as a tool to analyze the causal relationships between several exposures and SARS-CoV-2 spread. To get an overview on published associations, a scoping review was conducted from 20th to 22nd of May 2020 within Pubmed and Google scholar. Restrictions were applied to English and German language and the publication date in the last one year. The following search terms were applied to abstracts and title in Pubmed ("COVID-19" OR "COVID19" OR "Corona" OR "Coronavirus" OR "SARS-CoV-2") and connected separately in each case with the exposure variables ("mobility", "public awareness", "awareness", "google trends","ambient temperature", "temperature"). For "mobility", we analyzed $n = 8$ studies, $N = 103$ were scanned in Pubmed, together with the first ten pages (100 results) in Google scholar ("awareness"/"public awareness"/"google trends" $n = 9$, $N = 215$; "temperature"/"ambient temperature" $n = 16$, $N = 235$). We integrated these findings where possible into the construction of our DAG, which can be seen in Fig 1.

A number of studies report a strong association of **mobility** restrictions on the number of new COVID-19 cases: Restrictive measures (e.g. "stay-at-home" orders, travel bans, or school closures) are shown to possibly reduce the COVID-19 incidence [2, 15–21]. However, some studies point out the combination of various non-pharmaceutical interventions (NPIs) is decisive to prevent new infections [22, 23].

Google Trends [24] data can be used as a tool to get insights into public interest (**awareness**) in the coronavirus disease. Several recent studies imply a connection of relative search volumes (RSV) indices and reported new COVID-19 cases [4–6, 25–30]. Some search terms e.g. "COVID-19" or "coronavirus" predated newly infected cases/total number of cases by roughly 7 to 14 days for different countries [4–6, 26]. Additionally, we acknowledged that

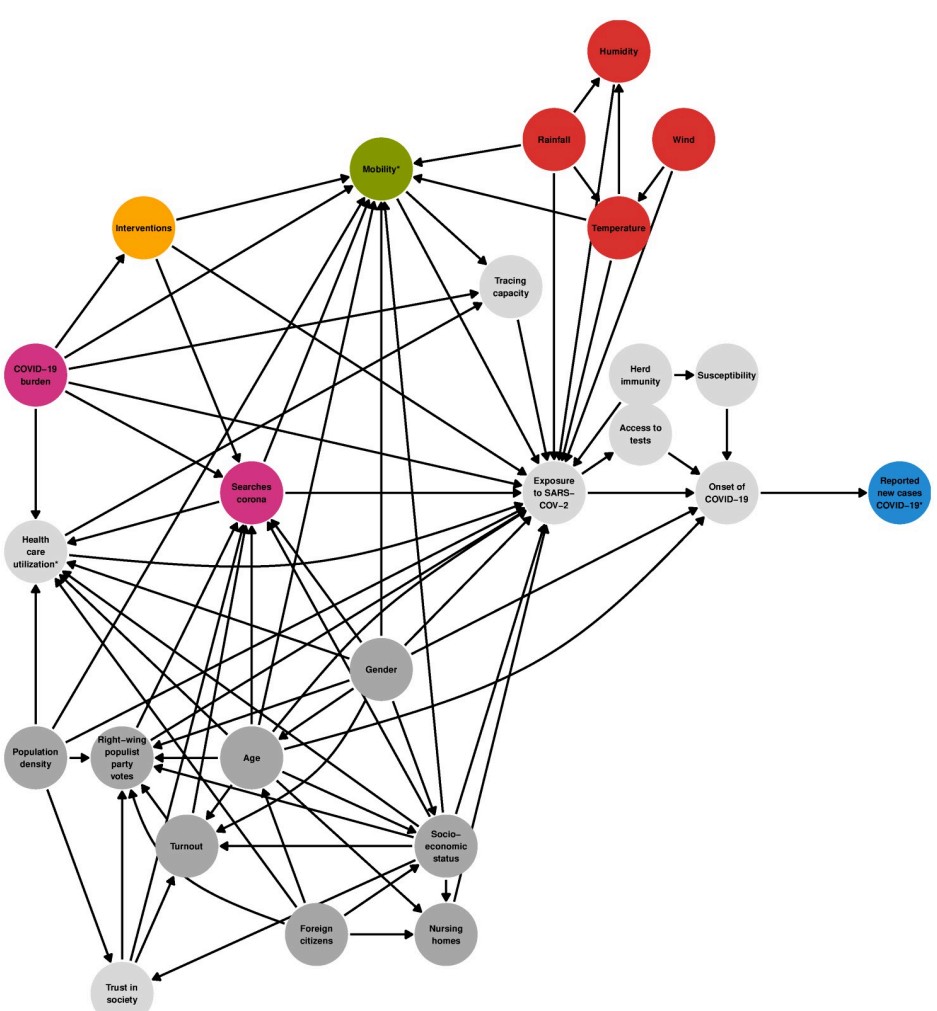

**Fig 1. DAG of determinants of reported COVID-19 cases on the district level.** Unobserved variables are light gray, variables marked with an asterisk (*) are confounded by weekday/holiday.

individual risk-aware behavior might be a reaction to the current COVID-19 burden (measured as reported cases at the day of exposure).

Mixed evidence is available regarding the effect of **temperature**: On the one hand several papers report an association between increase in temperature and decrease in newly infected COVID-19 cases [7–9, 31–36]. On the other hand, also the opposite has been found [37, 38]. Some studies found no association at all [22, 39–42]. It should be noted that few studies considered other confounding variables than meteorological ones (especially age and population density among others [22, 36, 39]). In addition, the transferability of results between different climate zones is questionable. To avoid possible bias caused by weather variables other than temperature, we included rain, wind, and humidity in our model.

When investigating causal determinants of SARS-CoV-2 infections, a number of confounders have to be considered. Well-known risk factors for SARS-CoV-2 as well as for other infections are demographic factors such as age, gender, socio-economic status (SES), population density, and foreign citizenship/ethnicity [13, 43, 44]. In Germany along with other countries (i.e. Brazil, USA, or the UK), populist parties or politicians and their electorate tend to be more

sceptical about effects of containment measures than the other part of the electorate [45, 46]. Therefore we considered both "right-wing populist party votes" and "voter turnout" as possible confounders. Public health interventions were also taken into account (contact restrictions, school closures etc.), as their implementation showed strong correlations with controlling the spread of SARS-CoV-2 [22, 23, 47]. To avoid bias due to reporting delay of case numbers we had to include weekday and German holidays. We included some unobserved variables in our DAG (e.g. "Herd immunity"), too. Please note that "Exposure to SARS-CoV-2" is itself an unobserved variable: German case numbers are reported with delay after date of exposure and symptom onset. *Exposure to the virus* should not be confused with the formal *exposure variables* of the DAG.

## Materials and methods

### Data

We collected and aggregated data on reported COVID-19 cases, regional socio-demographic factors, weather, and general mobility on district and state level in Germany for the period of 15 February 2020 to 8 July 2020. Our observation period for the outcome consisted of all dates from 20 February 2020 to 8 July 2020 ($T$ = 140), since we used a lag of 5 days for all confounders. We did not exclude any states or districts ($K$ = 401). We analyzed the daily reported number of new cases as outcome ($K \cdot T$ = 56 140 observations). The set of possible predictors was derived from our causal DAG (see Table 1 and Fig 1). Due to modelling and data limitations, some of the predictors were unobserved or were modelled as a construct consisting of several variables. For our causal graph analysis, we computed adjustment sets separately for all observed exposures within the DAG (if the respective exposure was identifiable within the DAG causal analysis framework).

**Variables.** We downloaded German daily case numbers on district level reported by Robert Koch Institute (RKI, [48]) and aggregated them by date. The number of daily active cases for day $d$ was derived by subtracting the total number of reported cases on day $d$ and day $d$ − 14 (14 days as a conservative estimate for the infectious period, which corresponds here to the required quarantine time in Germany).

To assess the mobility of the German population, we used data publicly available on German state level from Google [49]. Measurements are daily relative changes of mobility in percent compared to the period of 3 January 2020 to 6 February 2020. Missing values (25 out of 13 488) were imputed with value 0 and the state level measurements were passed onto districts within the corresponding state. Google mobility data was available for six different sectors of daily life ("retail and recreation", "grocery and pharmacy", "parks", "transit stations", "workplaces", "residential") which means that "mobility" is a construct consisting of several variables. All variables but "residential" mobility are relative changes of daily visitor numbers to the corresponding sectors compared to the reference period. "Residential" mobility is the relative change of daily time spent at residential areas. The six mobility variables showed high correlations among each other and with other variables. To reduce multicollinearity, we transformed them by principal component analysis (PCA) into six uncorrelated principal components which were used in place of the original variables.

The notion of awareness in the population of COVID-19 describes the general state of alertness about the new infectious disease. As such, it was hard to measure directly. As a proxy, we used the relative interest in the topic term "corona" as indicated by Google searches. The daily data was available on state level [24] and passed onto district level. As a second proxy for awareness, we used the daily reported number of COVID-19 cases on the day of the exposure:

**Table 1. Observed model variables.**

| Variable | Dynamics | Level | Type | Unit/comment | Source |
|---|---|---|---|---|---|
| Weekday | daily | national | categorical | Sat through Thu as six binary variables, Fri as baseline | - |
| Holiday (report) | daily | national | binary | - | - |
| Holiday (exposure) | daily | national | binary | - | - |
| **Mobility** | | | | | |
| Retail and recreation | daily | state | numeric | percent change compared to reference period | Google [49] |
| Grocery and pharmacy | daily | state | numeric | percent change compared to reference period | Google [49] |
| Parks | daily | state | numeric | percent change compared to reference period | Google [49] |
| Workplaces | daily | state | numeric | percent change compared to reference period | Google [49] |
| Residential | daily | state | numeric | percent change compared to reference period | Google [49] |
| Transit stations | daily | state | numeric | percent change compared to reference period | Google [49] |
| **Awareness** | | | | | |
| Searches corona | daily | state | numeric | percent relative to other states and observation period | Google [24] |
| COVID-19 burden | daily | district | numeric | reported cases on day of exposure | RKI [48] |
| **Weather** | | | | | |
| Rainfall | daily | district | numeric | mm (l/sqm) | DWD [50] |
| Temperature | daily | district | numeric | ˚C | DWD [50] |
| Humidity | daily | district | numeric | relative humidity (%) | DWD [50] |
| Wind | daily | district | numeric | m/s | DWD [50] |
| **Interventions** | | | | | |
| Ban of mass gatherings | daily | national | binary | - | - |
| School and kindergarten closures | daily | state | numeric | 0 for no closure, 1 for full closure, 0.5 for partial reopening | - |
| Contact restrictions | daily | national | binary | - | - |
| Mandatory face masks | daily | district | binary | - | IZA [52] |
| **Socio-demographic** | | | | | |
| Age | constant | district | numeric | 2 variables: share of population $>=65$ years & $<18$ years | INKAR [51] |
| Gender | constant | district | numeric | share of female population | INKAR [51] |
| Population density | constant | district | numeric | population per sqkm | INKAR [51] |
| Foreign citizens | constant | district | numeric | 2 variables: share of foreign citizens & of population seeking refuge | INKAR [51] |
| Socio-economic status | constant | district | numeric | share of households with low income | INKAR [51] |
| Turnout | constant | district | numeric | voter turnout in last election | INKAR [51] |
| Right-wing populist party votes | constant | district | numeric | share of votes for AfD in last election | INKAR [51] |
| Nursing homes | constant | district | numeric | number of nursing (retirement) homes | INKAR [51] |
| **Case numbers** | | | | | |
| Reported new cases of COVID-19 | daily | district | numeric | - | RKI [48] |
| Active cases | daily | district | numeric | active cases on day of report | RKI [48] |

Since media reported case numbers prominently, we assumed that this could reflect individual awareness, too.

We constructed daily weather from four variables ("temperature", "rainfall", "humidity", "wind"). Weather data was downloaded from Deutscher Wetterdienst (DWD, [50]) for all weather stations in Germany below 1000 meters altitude with daily records for our observation period. District level daily weather data was aggregated per district by averaging the data from the three nearest weather stations (which includes weather stations inside the district). Missing values were imputed with mean values ($n = 59$ for wind).

The reported number of COVID-19 cases varied strongly by day of the week. Thus, we included "weekday" as a categorical variable. Similarly, the reported cases and the exposure to the virus were affected by official holidays. Within the observation period, this included

among others Good Friday, Easter Monday, and Labor Day. To correct for effects of these days, we included two variables in the model, "Holiday (report)" (indicates if the day of the report was a holiday, because governmental health departments were less likely to be on full duty) and "Holiday (exposure)" (indicates if the day of exposure to the virus was a holiday, because the population behaves differently on holidays).

For different official and political interventions on a daily basis and the district level we used one-hot encoded daily variables, i.e. ban of mass gatherings, school and kindergarten closures and their gradual reopening, contact restrictions, and mandatory face masks for shopping and public transport.

We included several social, economic, and demographic factors on the district level with direct or indirect influence on the risk of exposure to SARS-CoV-2 in our analysis. All are readily available from INKAR database [51]. We used the share of population that is 65 years or older and the share of population that is younger than 18 years (Age), the share of females in population (Gender), the population density, the share of foreign citizenships and the share of the population seeking refuge (Foreign citizenship), the share of low-income households (Socio-economic status), voter turnout, share of right-wing populist party votes, and the number of nursing (retirement) homes.

All continuous variables but the outcome "Reported new cases of COVID-19" and the offset "Active cases" were centered and scaled by one standard deviation for numerical stability, while we left binary variables as-is. After estimating the effects of variables, we re-scaled continuous variables' effects to their original scale. Additionally for mobility variables, we re-transformed the effects of the principal components to the original mobility variables. Furthermore, we lagged the effect of all variables (but outcome, offset, and the non-dynamic socio-demographic variables) by 5 days (optimal lag found by cross-validation) which means that we assumed that their effects on the outcome will be visible after 5 days.

## Methods

**Causal analysis with DAG and adjustment sets.**   We used a directed acyclic graph as a graphical representation of the hypothesized causal reasoning that leads to exposure to the SARS-CoV-2 virus, onset of COVID-19, and finally reports of COVID-19 cases. We use the terms "causal effect" or "causal relationship" for effect estimates that are based on this causal graph framework. Every node $v_i$ in the graph is the graphical representation of an observed or unobserved variable $x_i$, a directed edge $e_{ij}$ is an arrow from node $v_i$ to $v_j$ that implies a direct causal relationship from variable $x_i$ onto variable $x_j$. The set of all nodes is denoted by $V$, the set of all edges by $E$, as such, the complete DAG is the tuple $G = (V, E)$. The seminal works of Spirtes and Pearl [53, 54] introduce the theory of causal analysis, do-calculus, and how to analyze a DAG to estimate the total or direct causal effect from a variable $x_i$ onto a variable $x_j$. The direct effect is the effect associated with the edge $e_{ij}$ only (if it exists), while the total effect takes indirect effects via other paths from $v_i$ to $v_j$ into account, too. Here we estimated total effects only, since most of our variables were not hypothesized to have a direct effect on the *reported* number of new COVID-19 cases. In contrast to prediction tasks, where one would include all variables available, it is actually ill-advised to use all available variables to estimate causal effects, due to introducing bias by adjusting for unnecessary variables within the causal DAG. This is why we need to identify a valid set of necessary variables (an adjustment set) to estimate the proper causal effect [54]. The "minimal adjustment set" [55] is a valid adjustment set of variables that does not contain another valid adjustment set as a subset. However, identifying a minimal adjustment set might not be enough to reliably estimate the causal effect. Thus, we

identified the "optimal adjustment set" [56] as the set of variables which is a valid adjustment set while having the lowest Akaike information criterion (AIC).

We analyzed the DAG from Fig 1 with the R Software [57] and the R packages `dagitty` (formal representation of the graph and minimal adjustment sets [12]) and `pcalg` (for finding an optimal adjustment set [58]). For the defined exposures and the outcome "Reported new cases of COVID-19", we computed the minimal and optimal adjustment sets. Since it was possible that these sets contained unobserved variables that needed to be left out of the regression model, we chose the valid set with the lowest AIC (see next section) to estimate the final total causal effect from exposure to outcome.

**Regression with negative binomial model.** We can estimate the causal effect from exposure to outcome by regression [54]. Since the outcome "Reported new cases of COVID-19" is a count variable, one should not employ a linear regression model with Gaussian errors, but instead we assumed a log-linear relationship between the expected value of the outcome $Y$ (new cases) and regressors $x$, as well as a Poisson or negative binomial distribution for $Y$:

$$\log(\mathbb{E}[Y|x]) = \alpha + \sum_{i \in S} \beta_i \cdot x_i, \tag{1}$$

where $\alpha$ is the regression intercept, $S$ is the set of adjustment variables for the exposure $i^*$ including the exposure variable itself, $\beta_i$ are the regression coefficients corresponding to the variables $x_i$. As such $\beta_{i^*}$ is the total causal effect from exposure variable $x_{i^*}$ on the outcome Y.

The Poisson regression assumes equality of mean and variance. If this is not the case one observes so-called overdispersion (the variance is higher than the mean), this indicates one should use regression with a negative binomial distribution instead to estimate the variance parameter separately from the mean.

We needed to account for the fact that our outcome is not counted per time unit (one day) only, but depends on the number of active COVID-19 cases: Holding all other variables fixed, the number of new cases $Y$ is a constant proportion of the number of active cases $A$. This was modeled by including an offset $\log(A + 1)$ in the regression model Eq (1):

$$\log(\mathbb{E}[Y|x]) \quad = \alpha + \log(A + 1) + \sum_{i \in S} \beta_i \cdot x_i$$

$$\Leftrightarrow \log\left(\frac{\mathbb{E}[Y|x]}{A + 1}\right) = \alpha + \sum \beta_i \cdot x_i \tag{2}$$

$$\Leftrightarrow \frac{\mathbb{E}[Y|x]}{A + 1} = \exp(\alpha) \cdot \prod \exp(\beta_i)^{x_i}. \tag{3}$$

Here we added a pseudocount "+1" to ensure a finite logarithm and avoid division by 0.

One can interpret the model as approximating the log-ratio of new cases and active cases by a linear combination of the regressor variables in Eq (2). If all variables $x_i$ are centered in Eq (3), we have for the baseline $\forall i \; x_i = 0 \Rightarrow E[Y|x = 0] = \exp(\alpha) \cdot (A + 1)$. In other words, the exponentiated intercept is the baseline daily infection rate (how many people does one infected individual infect in one day). If we hold all variables $x_i$ fixed (e.g. at baseline 0) in Eq (3) but now increase the exposure variable $x_{i^*} = 0$ by one unit to $x_{i^*} + 1 = 0 + 1$, we have

$$\mathbb{E}[Y|x'] \quad = \exp(\alpha) \cdot (A + 1) \cdot \exp(\beta_{i^*}^{x_{i^*}+1}) \prod_{i \neq i^*} \exp(\beta_i)^0$$

$$= \exp(\alpha) \cdot (A + 1) \cdot \exp(\beta_{i^*}),$$

which means the exponentiated coefficient $\beta_{i^*}$ describes the rate change of the outcome by one unit increase of the exposure.

In practice, given observations of $Y$ and $x$ we estimate the regression coefficients $\alpha$ and $\beta_i$ by maximum likelihood [59]. Our observational measurements are $y_{kt}$ and $x_{ikt}$, where $k$ indicates the corresponding district and $t$ the date of measurement.

We conducted a log-linear regression (function `glm` with `family = poisson()` for Poisson regression, and `glm.nb` from the `MASS` package for the negative binomial regression [60]) for the full data set to assess general model adequacy and to estimate the $\theta$ parameter of the negative binomial. The proper lag between exposures and outcome was found by 10-fold cross-validation on different lags between 1 and 20 days. Model diagnostics on the final full model did not show severe problems with model assumptions (linearity, distribution of residuals, independence of observations). Analysis of variance inflation factors revealed some problems with multicollinearity. To reduce the effects of multicollinearity, first we transformed the highly correlated mobility variables by PCA as described above. Second, we used a ridge regression approach [61], which is a regularization method that shrinks regression coefficients and alleviates the effect of correlation between variables on their respective regression coefficients. Furthermore, regularized regression allows for better fits on unseen data, thus preventing overfitting the data, too. The hyper-parameter $\lambda$ of the ridge regression was chosen by 10-fold cross-validation, where the folds were constructed from random subsets of the 401 districts. We used this hyper-parameter with the `cv.glmnet` function from the R package `glmnet` [62] with `family = negative.binomial(theta)` and chose the $\lambda$ value within one standard deviation from the minimal $\lambda$ as regularization hyper-parameter. Afterwards, we calculated the effects of separate exposures on the outcome. For every exposure, we analyzed the different valid adjustment sets given by analysis of the causal DAG (i.e. the minimal and optimal adjustment sets). Then, we first checked if the respective set included unobserved variables. If this was the case for the optimal adjustment set, we discarded the unobserved variables from the set and checked if it was still a valid adjustment set (function `gac` in package `pcalg` [63]). If a minimal adjustment set contained unobserved variables, we discarded the whole set. If no valid adjustment set for a given exposure was available, we concluded that the effect of this exposure was unidentifiable within our causal graph. We used the function `glmnet` with the parameters $\theta$ and $\lambda$ as above on every remaining valid adjustment set as regressors (that is, we applied ridge regression) and calculated the Akaike information criterion (AIC) for this model/set of regressors. Finally, for every exposure, we decided for the model/adjustment set (if available) with the lowest AIC. We report the exponentiated estimated coefficients for the separate exposures on their original scale.

## Results

Descriptive statistics for the included variables are presented in Table 2.

In the observational period, the number of daily reported COVID-19 cases increased till the end of March/beginning of April and continually decreased afterwards till the beginning of June 2020 with a slight increase and decrease afterwards (Fig 2A). On the other hand, the (log-)ratio of reported cases over active cases decreased steeply till the mid of April and increased steadily afterwards with a slight decrease close to the end of the observation period (Fig 2B). Both figures examplify a considerable variation among the districts (light blue points are individual district's data).

In Germany, we observed a rebound in mobility after the initial political measures, reductions in incident cases were associated with a diminishing public interest in COVID-19, and temperatures were overall increasing (cf. Fig 3); with correlations between temporal

**Table 2. Descriptive statistics for observed variables.**

| Variable | mean (SD) |
|---|---|
| n | 56140 |
| **Mobility** | |
| Retail and recreation | -26.62 (24.60) |
| Grocery and pharmacy | -3.94 (22.77) |
| Parks | 47.26 (58.20) |
| Workplaces | -22.96 (20.35) |
| Residential | 8.13 (6.49) |
| Transit stations | -29.58 (21.11) |
| **Awareness** | |
| Searches corona | 26.94 (18.23) |
| COVID-19 burden | 3.50 (10.28) |
| **Weather** | |
| Rainfall | 1.89 (4.01) |
| Temperature | 10.90 (5.33) |
| Humidity | 67.81 (13.03) |
| Wind | 3.63 (1.66) |
| **Interventions** | |
| Ban of mass gatherings | 0.83 (0.38) |
| School and kindergarten closures | 0.54 (0.36) |
| Contact restrictions | 0.74 (0.44) |
| Mandatory face masks | 0.49 (0.50) |
| **Socio-demographic** | |
| Age (pop. 65 and older) | 22.09 (2.74) |
| Age (pop. younger 18) | 16.17 (1.25) |
| Gender | 50.59 (0.64) |
| Population density | 533.75 (701.84) |
| Foreign citizens | 10.03 (5.14) |
| Foreign citizens (refugees) | 1.88 (1.14) |
| Socio-economic status | 30.64 (6.02) |
| Turnout | 75.08 (3.79) |
| Right-wing populist party votes | 13.39 (5.32) |
| Nursing homes | 36.11 (30.69) |
| **Case numbers (Outcome and offset)** | |
| Reported new cases COVID-19 | 3.53 (10.29) |
| Active cases | 48.76 (120.86) |

progression and mobility in retail and recreation $r_{A,B} = 0.02$, awareness ("Searches corona") $r_{A,C} = -0.3$, and temperature $r_{A,D} = 0.8$.

## Main results

We list the results of our causal analysis for the effects of different exposure variables in Table 3. The estimates are multiplicative rates of increase/decrease for a one unit increase of the respective variable: Values above 1 lead to an increase, below 1 to a decrease of the infection rate. To put these estimates into perspective, Fig 4 shows the relative causal effect of the different exposure variables on the number of reported COVID-19 cases on a range of sensible values of the exposure variables (95 percent quantiles of data points).

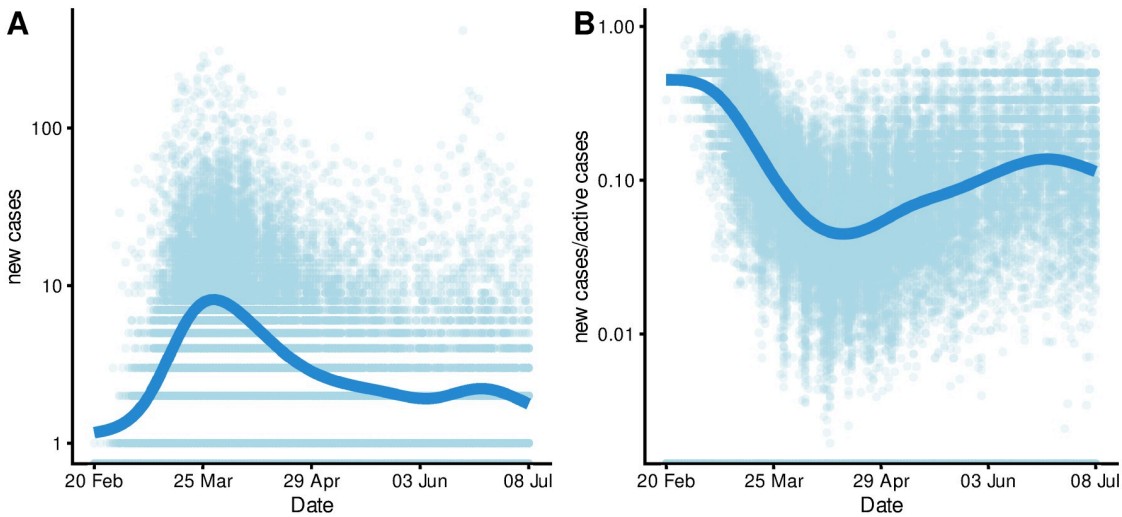

**Fig 2. Temporal and district level variation of outcome (log-scale).**

Within our framework, we saw very different effects for individual mobility variables. For mobility in retail/recreation, an increase of 1 percent point mobility compared to the reference period (03 January to 06 February 2020) leads to an increase of the daily reported case number by about 0.11 percent. Similarly, mobility on workplaces showed an effect of 0.33 increase in case numbers for every 1 percent point increase in mobility, while mobility on transit stations showed an effect of 0.26 increase in case numbers for every 1 percent point increase.

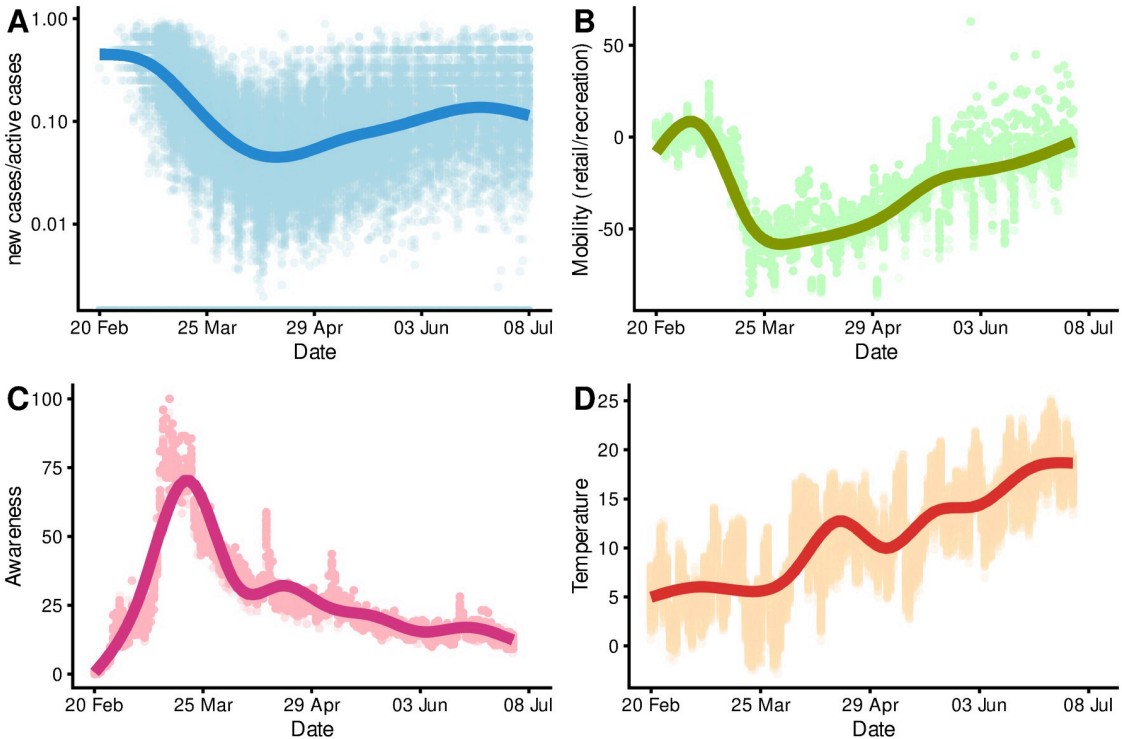

**Fig 3. Temporal variation of outcome and main determinants.**

**Table 3. Effect estimates from causal graph analysis.**

| Cause | Effect estimate |
|---|---|
| **Mobility** | |
| Retail and recreation | 1.0011 |
| Grocery and pharmacy | 0.9977 |
| Parks | 0.9997 |
| Transit stations | 1.0026 |
| Workplaces | 1.0033 |
| Residential | 0.9903 |
| **Awareness** | |
| Searches corona | 1.0089 |
| COVID-19 burden | 0.9980 |
| **Weather** | |
| Temperature | 0.9905 |
| Rainfall | 1.0121 |
| Humidity | 1.0057 |
| Wind | 1.0329 |
| **Interventions** | |
| Interventions (ban of mass gatherings) | 0.9729 |
| Interventions (school and kindergarten closures) | 0.9277 |
| Interventions (contact restrictions) | 0.8314 |
| Interventions (mandatory face masks) | 0.9064 |
| **Demographic** | |
| Age (pop. 65 and older) | 0.9953 |
| Age (pop. younger 18) | 1.0120 |
| Foreign citizens | 1.0048 |
| Foreign citizens (refugees) | 0.9985 |
| Gender | 0.9925 |
| Nursing homes | 1.0011 |
| Population density | 1.0000 |
| Socio-economic status | 0.9982 |

Contrarily, the remaining three mobility variables showed negative effects on the number of reported COVID-19 cases. An increase of 1 percent point mobility for the areas of grocery/pharmacy leads to a decrease in the reported case number by approximately 0.23 percent, while increased mobility of 1 percent point within parks leads to a decrease in the reported case number by approximately 0.03 percent, and finally an increase of 1 percent point in residential mobility leads to a decrease by approximately 0.97 percent. Fig 4 shows the effects of mobility on a range of possible values. Thus, we expect an increase of daily cases by approximately 7.8 percent if mobility in workplaces reaches baseline levels of 0 percent difference to the reference period. On the other hand, an increase of mobility for residential areas by 10 percent points compared to the reference period leads to a reduction of the infection rate by approximately 1.8 percent.

"Awareness" had two opposite effects on the outcome in our DAG. Awareness measured by Google searches for *corona* had a positive effect on the number of reported cases. An one percent point increase of the state's Google searches (relative to other states and the observation period) leads to an increase of approximately 0.89 percent. For example, if a district shows 10 percent points more relative searches for *corona* than another one, we expect approximately

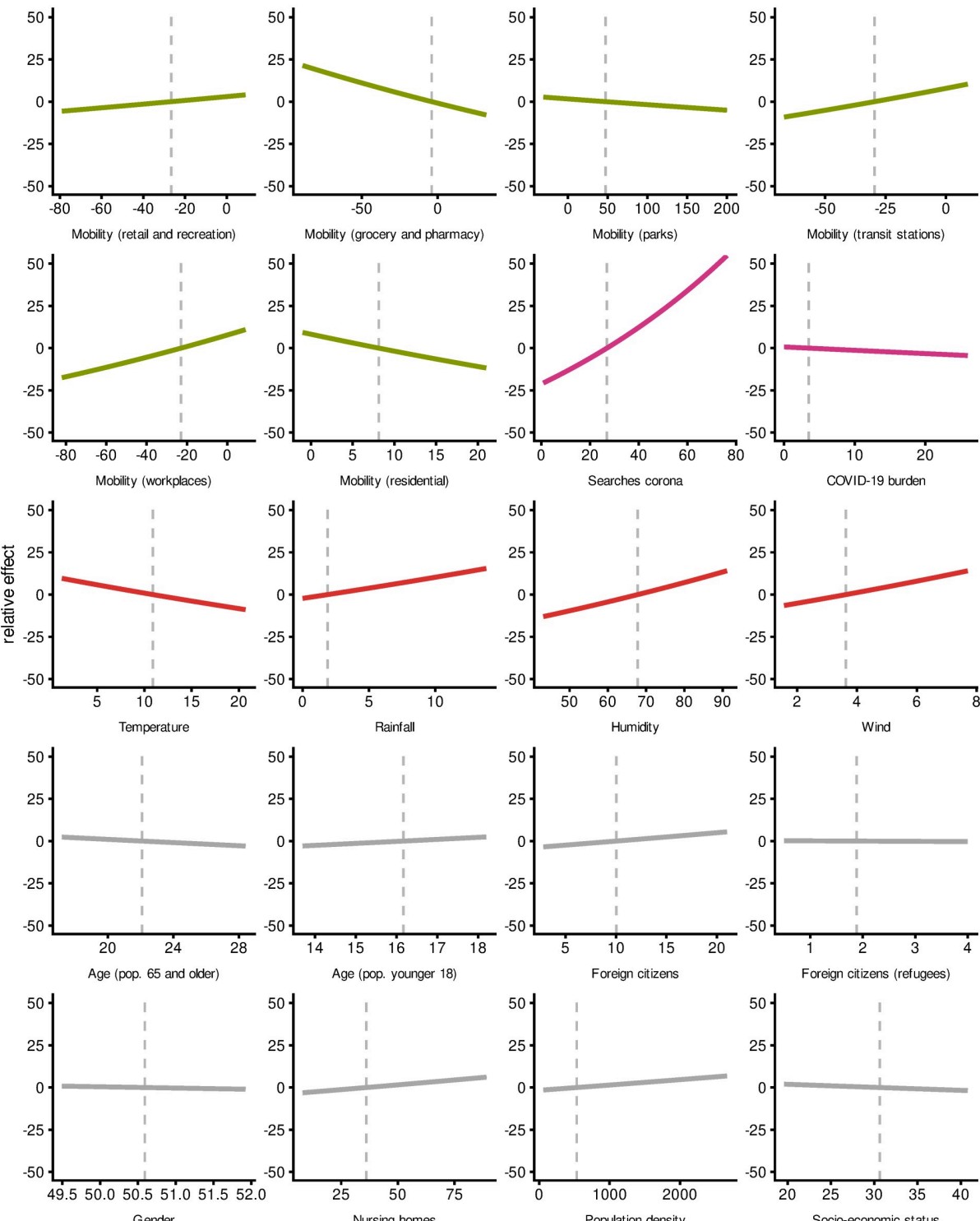

**Fig 4. Relative causal effects of exposures.**

9.3 percent more infections for this district after 5 days. *COVID-19 burden* (reported number of cases on day of exposure) affected the outcome negatively, where every additional daily case in the district leads to a 0.2 percent decrease in newly reported case numbers. The corresponding plot in Fig 4 visualizes this relationship: For a local outbreak with 20 daily cases as COVID-19 burden, we estimate as total causal effect a subsequent reduction of infection rate by 3.9 percent.

Within our model, we observed effects of temperature and all other weather variables. Every increase of 1 degree Celsius in temperature leads to a reduction of the daily reported case numbers by approximately 0.95 percent. On the other hand, we found an increasing effect of rainfall: One millimeter (=1 liter per square meter) more rainfall leads to an increase of reported case numbers by approximately 1.21 percent. We observe effects for humidity and wind as well (higher humidity and stronger wind leading to more cases). In perspective (Fig 4), with temperature we expect an increase by approximately 21 percent at a daily average temperature of $0°C$ compared to a day with $20°C$. For rainfall, we expect on a rainy day with 10 mm rainfall a corresponding increase of the infection rate by approximately 12.8 percent compared to a day with no precipitation.

The different intervention variables showed the strongest effects in our analysis, see Table 3. While the first intervention (ban of mass gatherings) reduced subsequent daily case numbers by 2.7 percent, the closure of schools/kindergartens reduced infections by an additional 7.2 percent and mandatory face masks reduced this by another 9.4 percent. The effect of contact restrictions was the strongest in our observation period, with an reduction of the case rates by 16.9.

The effects of the different socio-demographic factors are quite small in comparison to the effects described above. We see an increasing effect on case numbers by additional nursing homes between districts. Districts with a younger population, more foreign citizens, higher population density and a lower average social-economic status showed higher case numbers, too.

For all exposures, our analysis pipeline opted to use the (reduced) optimal adjustment set over the minimal adjustment sets because of lower AICs, except for exposure variable "nursing homes", for which the minimal adjustment set had the lowest AIC. For an overview of all final adjustments sets, see Table 4. We found that there were no valid adjustment sets for the non-identifiable variables turnout and right-wing populist party votes.

We decided for a lag of 5 days based on cross-validation. Similarly, negative binomial regression was chosen over Poisson regression, because the latter showed overdispersion and an higher AIC value.

## Discussion

### Main findings

Our objective was to identify effects of determining factors for COVID-19 cases within a causal framework. We found that weather affects the reported number of infections, especially temperature (which has a reducing effect on case numbers) and rainfall (which increases case numbers). We saw that reports of high case numbers in districts led to a reduction in new infection numbers, which indicates risk-averse awareness in the population and/or effective public health measures to suppress a local outbreak. Mobility showed distinct effects: Increasing activity in retail and recreational areas, as well as transit stations and workplaces increased reported case numbers, while increased movement for essential shopping (grocery and pharmacy) and in parks or residential areas led to reduced case numbers. All interventions considered (ban of mass gatherings, school/kindergarten closures, contact restrictions, mandatory

**Table 4. Final adjustment sets for causal analysis.**

| | Mobility | Searches corona | COVID-19 burden | Temperature | Rainfall | Humidity | Wind | Interventions | Age | Foreign citizens | Gender | Nursing homes | Population density | Socio-economic status |
|---|---|---|---|---|---|---|---|---|---|---|---|---|---|---|
| Weekday | x | x | x | x | x | x | x | x | x | x | x | x | x | x |
| Holiday (report) | x | x | x | x | x | x | x | x | x | x | x | x | x | x |
| Holiday (exposure) | x | x | x | x | x | x | x | x | x | x | x | x | x | x |
| **Mobility** | | | | | | | | | | | | | | |
| Mobility | | | | | | x | | | | | | | | |
| **Awareness** | | | | | | | | | | | | | | |
| Searches corona | x | | | x | x | x | x | | | | | x | | |
| COVID-19 burden | x | | | x | x | x | x | x | x | x | x | x | x | x |
| **Weather** | | | | | | | | | | | | | | |
| Temperature | x | x | x | | | x | | x | x | x | x | | x | x |
| Rainfall | x | x | x | x | | x | x | x | x | x | x | | x | x |
| Humidity | x | x | x | | | | | x | x | x | x | | x | x |
| Wind | x | x | x | x | x | x | | x | x | x | x | | x | x |
| **Interventions** | | | | | | | | | | | | | | |
| Interventions | x | x | | x | x | x | x | | | x | x | x | x | x |
| **Socio-demographic** | | | | | | | | | | | | | | |
| Age | x | x | x | x | x | x | x | x | | | x | x | x | x |
| Gender | x | x | x | x | x | x | x | x | x | x | | x | x | x |
| Population density | x | x | x | x | x | x | x | x | x | x | x | x | | x |
| Foreign citizens | | x | x | x | x | x | x | x | x | | x | | x | x |
| Socio-economic status | x | x | x | x | x | x | x | x | | | | x | x | |
| Turnout | | | x | | | | | x | | | | | | |
| Right-wing populist party votes | x | x | x | x | x | x | x | x | | | | x | | |
| Nursing homes | x | x | x | x | x | x | x | x | | | | | x | |

face masks) reduced case numbers considerably. Socio-demographic variables had small effects individually, but in conjunction they explained larger case numbers in (urban) areas with younger population, lower socio-economic status, and higher population density.

Furthermore, we made a strong case for the use of causal DAGs in epidemiology and a pandemic like COVID-19: DAGs allow to choose confounders for the analysis in a principled and statistically correct way while reducing possible causes for bias. Also, the DAG formalization allows for discussion about the underlying causal assumptions.

## Comparison with previous research

Most research on determinants affecting case numbers of COVID-19 is restricted to single aspects [5, 16, 32, 35]. To reliably identify (causal) drivers, one must adjust for confounders. To this end, we used an integrated model with variables from different aspects like mobility, awareness, weather, or socio-demographics and identified confounders by causal analysis with a directed acyclic graph. A causal approach is used in another current COVID-19 analysis [64]. There, however, they identify the causal relationships (reconstruct a DAG), while we estimated effects for a given hypothesized causal DAG.

Several studies assessing the impact of public health measures on mobility have each observed a downward trend accompanied by a decrease in the number of newly reported cases [15–17, 19, 23, 47].

Our findings regarding awareness/Google Trends analysis are in good agreement with the correlations found by others [4, 6, 26], who conclude that alertness to COVID-19 rises several days before the highest number of cases are reported. At this point it should be noted, that awareness is substantially influenced by public media coverage, which should be considered, if possible, in future studies [4]. As such, awareness is difficult to measure and here the number of Google searches for "corona" could only be a proxy for this concept.

In addition, in alignment with other recent published studies, our results confirm evidence which associated a negative effect of temperature on new COVID-19 cases [7–9, 31–36]. It is however controversial to other scientific literature describing no effects [22, 39–42] or even converse correlations [37, 38]. The conflicting results might be explained by different climates and characteristics of the populations under study. While we are confident that our strict causal analysis resulted in effect estimates as undistorted as possible, there might be unconsidered bias in those other studies. Further research needs to be done to elucidate the biological characteristics of the novel virus SARS-CoV-2 regarding its ambient temperature survival and transmission. Finally, we found a positive effect of increased precipitation and a raise in COVID-19 cases, which supports previous observations [33].

A recent review on COVID-19 based on evidence from the US and UK concludes that low socio-economic status groups are being hit harder by the pandemic [65]. Albeit specific pathways remain unclear, many studies found associations with poverty or its correlates such as poor and potentially overcrowded housing conditions. For Germany, a higher case fatality of COVID-19 cases in districts with higher socio-economic deprivation has also been reported just recently, which was especially pronounced in the second wave of the pandemic [66]. Similarly, our analysis identified a decreasing effect on COVID-19 case numbers within districts with a higher socio-economic status during the first wave.

## Limitations and strengths

While use of a causal DAG is itself a strong tool to identify *causal* effects (and not just statistical associations), it introduces two limitations: causal assumptions within the graph (depicted by edges) need to be well justified, and the statistical regression model that calculates total causal

effects needs to be appropriate for the task at hand. We endorse our graph as a basis for discussion on residual confounding. We did not try to construct the DAG from the available data (cf. [64]). As such, our proposed DAG is not entirely consistent with the data and there are conditional dependencies between variables that cannot be dissolved by adding edges to the DAG (e.g. between the interventions like contact restrictions and mandatory face masks). Another way to identify potential problems in the proposed DAG is to perform a sensitivity analysis of its structure by inspecting its maximal ancestral graph (MAG) or its Markov equivalence class represented by a complete partially DAG (CPDAG) and the existence of valid adjustment sets for these generalized graphs [67]. For the MAG derived from our DAG, only the effects for exposures mobility and searches for corona can be estimated with valid adjustment sets, while for the Markov equivalence class all exposures but COVID-19 burden lead to valid adjustments sets. A further analysis of these implications is out of the scope of this paper.

We observed overdispersion and a substantial increase in model performance with a negative binomial regression compared to Poisson regression, which is in line with the results on COVID-19 daily case counts of [17] and others [7, 9, 68]. We did not model case counts with a differential equation model like the classic SIR-model [69] and its successors, since these are more suited to prediction e.g. [70], while our choice of a negative binomial regression framework allowed us to estimate the effects of confounders more reliably. There are more advanced statistical methods for count data, e.g. zero-inflated models and mixed models. We tested both approaches as extensions to the negative binomial regression and experienced numerical problems and increased computing time, along with an insubstantial increase in model performance. Furthermore, our model assumed that all variables have effects proportional to the size of their measurements. It is possible that some variables show saturation effects or opposite effects for low, medium, or high values. This could be modeled with polynomial or other transformations of the variables, which we did not employ due to limited temporal and spatial data availability. Interaction effects of variables and confounding effects or mediating variables are explicitly taken care of by deriving the valid adjustment sets for a given exposure based on the causal DAG. Use of a fixed DAG with effect estimation via regression assumes that data was generated by the same underlying process for the observation period. By inclusion of the successive mitigation interventions as binary variables we were able to explain some of the variance caused by the changing dynamics of case numbers (similar to [68]). While multicollinearity of variables poses less of a problem for a proper causal graph analysis [71], we addressed the problem of multicollinearity in our predictors by two approaches: principal component analysis for the highly collinear mobility variables as well as a regularized regression approach (ridge regression). The latter (in conjunction with cross-validation) also reduced the problem of overfitting.

We stress the point that our effects were deduced on an aggregate (district) level in the absence of available data on an individual level. As such, conclusions about effects cannot be transferred on individuals without the possibility for an ecological fallacy. Furthermore, as we were using administrative data for our analysis, the results are susceptible to the Modifiable Area Unit Problem (MAUP) [72]. The MAUP postulates that different regional aggregations of the units of observation may lead to different results and conclusions. Due to limited available data for the different variables, there is currently no way to overcome these problems that are inherent to all analyses on aggregated data level.

Our observation period was restricted to succession from late winter to spring and summer (February to July). Nevertheless, this transition with increasing temperature was a natural experiment that allowed clues on weather effects.

We could not include data on health care utilization during the pandemic into our models due to the lack of available resources. This is planned for a later follow up to this paper since

we rank health care utilization and mobility within health care facilities among the strong factors for COVID-19 progression: personnel in hospitals and private practices is particularly exposed to infection, while the lack of adequate care for other diseases has severe effects on general health of the population. At the same time, health care facilities are key for testing and surveillance of COVID-19 patients.

Social determinants of health are important factors to consider in an epidemiological framework of a pandemic disease like COVID-19. To account for this problem, we included several socio-economic confounders that were available on a district level in Germany. While our analysis is not an exhaustive analysis of the effects of social determinants on COVID-19 infections, we emphasize the necessity of their inclusion and our results add to the growing body of evidence that these factors interact with each other and cluster especially among people or within areas of underprivileged conditions, with detrimental effects on population health [73].

While our analysis focused on Germany and its districts, we assume that results may be transferred to other countries by adjusting for their respective weather conditions, mobility habits, socio-demographic characteristics, and other determining factors.

The code and resources for our analysis are available on Github, we invite other researchers to replicate our analysis with different assumptions using the files provided in the repository of the article (https://github.com/zidatalab/causalcovid19).

## Discussion of causal effects

In our analysis, the adverse effects of mobility in retail/recreation and workplaces and the favorable effect of mobility in grocery/pharmacy and residential areas indicate that interventions like contact restrictions which limit the number of individual interactions can lead to reduced infection numbers. This is due to retail/recreational and workplace areas encompassing mostly places of (social) gatherings, while if people are doing more of their essential shopping at supermarkets and stay at home with less contact to other people, they are less likely to come in contact with infected individuals.

The effects of awareness measured via searches for "corona" and the COVID-19 burden are harder to interpret. We assume that within our model, the searches for "corona" are an insufficient proxy for awareness, while the decreasing effect for future case numbers of high daily COVID-19 burden indicates it affects individual risk-behavior and entails effective non-pharmaceutical interventions.

Similarly, the effects of temperature and rainfall can be interpreted as causal effects for indoor and outdoor activities, such that higher temperatures and low rainfall indicate more people spending time outdoor while lower temperatures and high rainfall result in indoor activities, which lead to more infections. Current research suggests this to be due to the prevalent airborne and respiratory droplets and aerosol transmission of the SARS-CoV-2 virus [74]. In this light, we advocate for precautious measures like increased hygiene, face masks, and air ventilation for unavoidable indoor activities.

Furthermore, our analyses strongly support the effectiveness of non-pharmaceutical interventions. To a lesser extent, the adverse effects of some socio-demographic factors might help to identify areas that are at higher risk of local COVID-19 outbreaks and more severe outcomes of infection cases.

## Conclusion

To the best of our knowledge, this is the most comprehensive analysis of causes for COVID-19 infections which integrates different data sources (all publicly available). Causal reasoning with a DAG allows us to estimate the possible causal effects more reliably.

Our findings suggest that the infection-driving effects of mobility, awareness, and weather (and to some extent socio-demographic factors) need to be taken into account when deciding for mitigation and suppression interventions, depending on the recent and future COVID-19 pandemic development.

## Acknowledgments

We are thankful for feedback from Thomas Czihal, Johannes Textor, Ralph Brinks, and an anonymous reviewer who gave helpful suggestions on earlier versions of the manuscript.

## Author Contributions

**Conceptualization:** Edgar Steiger, Lars Eric Kroll.

**Data curation:** Edgar Steiger.

**Formal analysis:** Edgar Steiger.

**Investigation:** Edgar Steiger, Tobias Mussgnug.

**Methodology:** Edgar Steiger, Lars Eric Kroll.

**Project administration:** Lars Eric Kroll.

**Software:** Edgar Steiger.

**Supervision:** Lars Eric Kroll.

**Visualization:** Edgar Steiger.

**Writing – original draft:** Edgar Steiger, Tobias Mussgnug, Lars Eric Kroll.

**Writing – review & editing:** Edgar Steiger, Tobias Mussgnug, Lars Eric Kroll.

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
