## [Decision Letter · Decision Letter 0]

8 Feb 2021

PONE-D-20-23587

Causal analysis of COVID-19 observational data in German districts reveals effects of mobility, awareness, and temperature

PLOS ONE

Dear Dr. Steiger,

Thank you for submitting your manuscript to PLOS ONE. After careful consideration, we feel that it has merit but does not fully meet PLOS ONE’s publication criteria as it currently stands. Therefore, we invite you to submit a revised version of the manuscript that addresses the points raised during the review process.

We look forward to receiving your revised manuscript.

Kind regards,

Sungwoo Lim, DrPH

Academic Editor

PLOS ONE

Reviewers' comments:

Reviewer's Responses to Questions

**Comments to the Author**

1. Is the manuscript technically sound, and do the data support the conclusions?

Reviewer #1: Yes

2. Has the statistical analysis been performed appropriately and rigorously? 

Reviewer #1: No

3. Have the authors made all data underlying the findings in their manuscript fully available?

Reviewer #1: Yes

4. Is the manuscript presented in an intelligible fashion and written in standard English?

Reviewer #1: Yes

5. Review Comments to the Author

Reviewer #1: This is an interesting manuscript that aims to identify "causal" relationships" for COVID-19 transmission based on a DAG analysis and empirical data from other studies for variable selection.

The DAG analysis is a multivariate analysis to select a subset of variables that are correlated from a large number of candidates. The first select the sets of variables that are associated with the outcome using DAG analysis and then put them into the regression. The DAG analysis returns different sets of variables, using different selection criterion ( for example, the most parsimonious set or the most variation explained ). So the authors assumedly use R-squared to select the final sets of variables in the regression model.

The challenge I have here (and having reviewed this as well with our statistical team) is that the claims of causal relationships are not convincing, even if they identify associative relationships that other studies have also found.

Specifically:

1) The model does not address the nonlinearity of variables (such as temperature); other studies have found that temperature has a u-shaped effect at low and high temperatures. Humidity effects may also be nonlinear. The assumption of non-linearity on continuous variables needs to be considered.

2) There are no tests for multi-collinearity that are presented, which presents significant concerns for over-fitting the model. Temperature and humidity are an example here. So is the mobility data.

2) Feels like they cherry-pick the variables of interest in the end? Not sure how they arrived at the final variable list for causal effects. Whey were the restrictions not included in that analysis?

3) The term “causal analysis” is a bit strong for what they have done here. The basis of the work is the proposed DAG (Figure 1), but the diagram was constructed from other association studies. Other than the DAG analysis, they did not do anything to ensure the results are “causal relationship”. So I am not convinced that the analysis or results are causal.How do you avoid over-fitting the model or including mediating variables in the analysis?

5) What about interaction terms? For example, residential mobility and colder weather? Or rain? These relationships are not simply a straightforward multivariable model.

6) The most interesting terms in their model were the interventions (restrictions) themselves, yet they were dropped from the model. No discussion is really pursued on that. Why were public health interventions removed?

6. PLOS authors have the option to publish the peer review history of their article (what does this mean?). If published, this will include your full peer review and any attached files.

Reviewer #1: No

---

## [Author Response · Author response to Decision Letter 0]

29 Mar 2021

Our responses to the reviewer's comments are in the pdf Reply to the Reviewer.

Fell free to contact us if you need additional information.

---

## [Decision Letter · Decision Letter 1]

13 Apr 2021

PONE-D-20-23587R1

Causal graph analysis of COVID-19 observational data in German districts reveals effects of determining factors on reported case numbers

PLOS ONE

Dear Dr. Steiger,

Thank you for submitting your manuscript to PLOS ONE. After careful consideration, we feel that it has merit but does not fully meet PLOS ONE’s publication criteria as it currently stands. Therefore, we invite you to submit a revised version of the manuscript that addresses the points raised during the review process.

We look forward to receiving your revised manuscript.

Kind regards,

Sungwoo Lim, DrPH

Academic Editor

PLOS ONE

Journal Requirements:

Reviewers' comments:

Reviewer's Responses to Questions

**Comments to the Author**

1. If the authors have adequately addressed your comments raised in a previous round of review and you feel that this manuscript is now acceptable for publication, you may indicate that here to bypass the “Comments to the Author” section, enter your conflict of interest statement in the “Confidential to Editor” section, and submit your "Accept" recommendation.

Reviewer #2: (No Response)

2. Is the manuscript technically sound, and do the data support the conclusions?

Reviewer #2: Yes

3. Has the statistical analysis been performed appropriately and rigorously? 

Reviewer #2: Yes

4. Have the authors made all data underlying the findings in their manuscript fully available?

Reviewer #2: Yes

5. Is the manuscript presented in an intelligible fashion and written in standard English?

Reviewer #2: Yes

6. Review Comments to the Author

Reviewer #2: I find the topic of the paper very important and the methodology very interesting. The author has used a number of variables to analyse possible determinants of COVID-19 infections. While the paper acknowledges that there are social factors which affect the spread of the COVID-19 this is not properly discussed in the paper. I strongly suggest the author to refer to the literature on Social Determinants of Health when discussing social variables. For example see the recent paper by Galanis and Hanieh in Social Science and Medicine on Incorporating Social Determinants of Health into Modelling of COVID-19 and references therein.

7. PLOS authors have the option to publish the peer review history of their article (what does this mean?). If published, this will include your full peer review and any attached files.

Reviewer #2: No

---

## [Editor Report · Decision Letter 2]

6 May 2021

Causal graph analysis of COVID-19 observational data in German districts reveals effects of determining factors on reported case numbers

PONE-D-20-23587R2

Dear Dr. Steiger,

We’re pleased to inform you that your manuscript has been judged scientifically suitable for publication and will be formally accepted for publication once it meets all outstanding technical requirements.

Kind regards,

Sungwoo Lim, DrPH

Academic Editor

PLOS ONE
---

## [Editor Report · Acceptance letter]

17 May 2021

PONE-D-20-23587R2 

Causal graph analysis of COVID-19 observational data in German districts reveals effects of determining factors on reported case numbers 

Dear Dr. Steiger:

I'm pleased to inform you that your manuscript has been deemed suitable for publication in PLOS ONE. Congratulations! Your manuscript is now with our production department. 

Kind regards, 

on behalf of

Dr. Sungwoo Lim 

Academic Editor

PLOS ONE